# Observing shifts in phenology of tropical flowering plants

Skylar Graves[1]*, Erin A. Manzitto-Tripp[1,2]

1 Ecology and Evolutionary Biology, University of Colorado-Boulder, Boulder, Colorado, United States of America, 2 Museum of Natural History, University of Colorado-Boulder, Boulder, Colorado, United States of America

* skly5321@colorado.edu

## Abstract

Changes in flowering can cause misalignment with pollinators and seed dispersers, thus causing changes in fitness of both the plant species and their mutualists. Phenological shifts in tropical flowering plants are poorly documented despite the widely understood importance of measuring flowering phenology. It has been hypothesized that changes in tropical flowering have been less severe than those in non-tropical latitudes due to minimal change in temperature annually. Furthermore, many tropical species flower continuously throughout the year, as they are not restricted by a cold induced dormancy period. To test this hypothesis, we used museum specimens to examine shifts in phenology of flowering plant species from across the global tropics. We identified species that flower once a year, every year, for four consecutive months or less, as these species can best be compared to temperate studies. This selection criteria resulted in 33 species across the tropics. Between 1794 and 2024, we documented an average absolute shift in flowering of 2.04 days per decade across all species, with a range of 0.037 days per decade to 14.10 days per decade. This shift is comparable to changes seen elsewhere around the globe, including those in temperate, boreal and alpine desert plants. This change has been shown to be severe enough to cause interspecific misalignment. Our work shows that changes in tropical flowering phenology are not insulated from the impacts of climate change, as previously assumed.

## Introduction

Climate change has driven shifts in the behavior and life cycles of animals, plants, and many other organisms [1–3]. For plants, numerous studies in non-tropical habitats have documented the growing season lengthening, causing spring and autumn flowering to occur earlier and later, respectively [1–3]. Responses of plants to climate change can have cascading impacts across ecosystems [1,4–8]. For example, climate change has induced changes in insect emergence [9–14], which, coupled with

**Data availability statement:** All data and code used in this study are available at: https://github.com/SkylarMGraves/Observing-shifts-in-phenology-of-tropical-flowering-plants.

**Funding:** The author(s) received no specific funding for this work.

**Competing interests:** The authors have declared that no competing interests exist.

changes in flowering timing can cause pollinator misalignment. Furthermore, climate change can impact migration timing [15], which in turn can disrupt seed dispersal. These changes and more in turn fracture communities and food chains [5,6,16,17]. Plant phenology is especially sensitive to a changing climate, with changes in reproductive cycles serving as immediate and visible indicators of broader change and predictors of more drastic ecological responses [7,18,19]. Because plants are the basis of the vast majority of terrestrial ecosystems, understanding changes in plant phenology informs us of broader impacts of climate change.

Despite the universally recognized importance of our documentation of shifts in flowering phenology [1–20], tropical latitudes have been largely overlooked. Global tropical latitudes house approximately 57% of global vascular plant biodiversity [20] and 62% of global vertebrate biodiversity [21]. In addition, nearly 180 species of plants (and ~1,100 non-plant species) from the tropics are described as new to science each year [20]. Given the vast biodiversity seen in the tropics this is a large blind spot regarding understanding the global impacts of climate change. Furthermore, biodiversity increases the complexity of the ecosystem dynamics, and thus the effect misalignment in interspecific interactions can cause [20–21]. Due to the lack of cold-induced dormancy period and consistent photoperiod throughout the year [22], flowering patterns of many tropical plant species are drastically different from those of non-tropical species [22]. Many species have continuous flowering patterns, meaning they flower consistently throughout most or all of the year. However, other species have discrete flowering patterns that more closely mirror those in non-tropical and boreal regions [22]. Species that have discrete flowering periods are more tractable from a perspective of understanding phenological change, as these flowering patterns mirror those seen in non-tropical latitudes. Such species therefore enable comparison to phenological changes documented from non-tropical latitudes. From an ecological perspective, species with discrete flowering periods have a higher likelihood of misaligned interspecific interactions as a result of changes in reproductive period [22,23]. Measuring species with discrete flowering times is therefore likely to target the species most vulnerable to change.

Ecological changes that affect tropical latitudes have cascade impacts across the planet, such as changes in nutrient cycling which can cause changes in atmospheric $CO_2$ uptake globally and alter nutrient runoff which effects global marine ecosystems [22]. As a result, documentation of phenological shifts in and amongst tropical floras is a crucial piece of information towards more complete understanding of the global impacts of anthropogenic climate change [24]. Here, we identified 33 species of flowering plants from tropical latitudes that have discrete flowering times, spanning 15 families and 8 locations, then asked whether (1) these species have undergone changes in flowering phenology through time, and (2) how these changes compared to those seen in plants from non-tropical latitudes.

## Methods

We selected tropical locations that were proximal to biological research stations and/or preserves, increasing the likelihood of steady collection efforts through time.

These locations (Table 1; Fig 1) were chosen specifically due to their large number of herbarium collections as well as the consistency of the collections throughout time [25–31]. The temporal regularity of collections minimizes collector bias inherent in the usage of museum specimens. These locations were subsequently pared down to fewer candidate sites after criteria for inclusion were applied (see below, Table 2). Rates of collections were observed. Across all locations there was a decrease in collections from 1914 to 1945, likely due to global events, but the consistency in collections before and after yielded a robust dataset.

### Herbarium data retrieval and dataset compilation

The use of herbarium specimens introduces the concern of bias in datasets. However, it has been shown that the unwanted impacts of collector bias can be ameliorated with large datasets and utilization of mean flowering, as opposed to peak or first flowering [25–30,32,33]. First flowering refers to the time when the earliest flowers open, whereas peak flowering refers to the time when the largest number of flowers are in bloom. Due to the nature of museum specimens, the

**Table 1. Approximate centroid latitude and longitude coordinates of locations included in this study.**

| Location | Latitude | Longitude |
| --- | --- | --- |
| INPA Reserves, Brazil | −3.119 | −60.021 |
| Jatun Sacha, Ecuador | −1.066 | −77.616 |
| Las Cruces, Costa Rica | 8.954 | −83.070 |
| Tropenbos International, Bolivia | −17.846 | −60.738 |
| Cocha Cashu, Peru | −11.888 | −71.407 |
| Catimbau National Park, Brazil | −8.592 | −37.247 |
| Isthmus of Kra, Myanmar | 10.333 | 99.000 |
| Bia National Park, Ghana | 6.504 | −3.077 |
| Southern Guinea Savanna Research Station, Guinea | 9.296 | 5.063 |

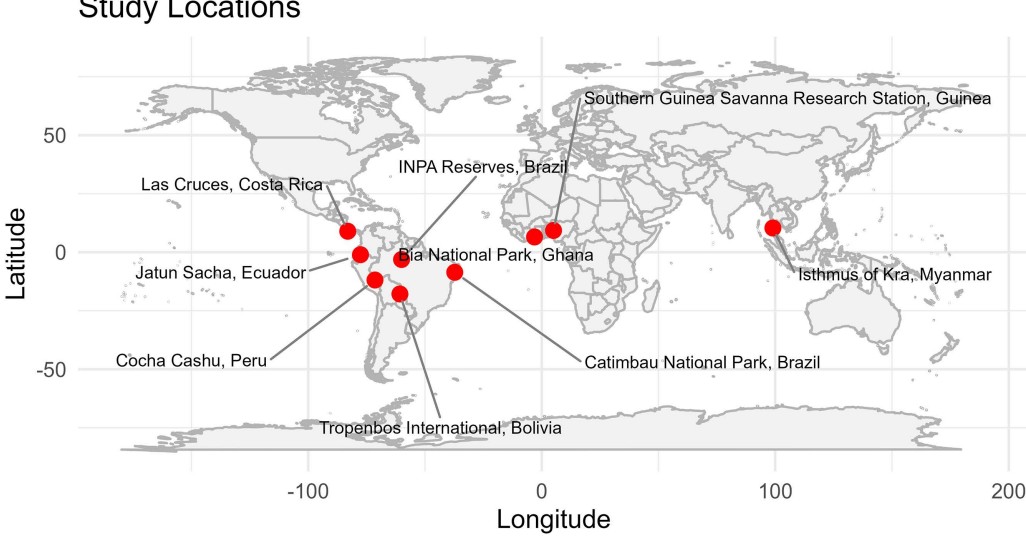

**Fig 1. Map of locations for the 33 species utilized in this study.** Map created using the R package "maps". INPA Reserves Amazon Basin, Brazil. Jatun Sacha, Ecuador. Tropenbos International, Bolivia. Catimbau National Park, Brazil. Cocha Cashu, Peru. Bia National Park, Ghana. Southern Guinea Savanna Research Station, Guinea. Isthmus of Kra, Thailand/ Myanmar.

**Table 2. Table includes location, species name and taxonomic authority, IUCN Red List Status, known pollinators, known seed dispersal agents and habit.**

| Location | Species | IUCN | Pollinator | Seed dispersal | Habit |
|---|---|---|---|---|---|
| INPA Reserves, Brazil | *Ceiba erianthos (Cav.) K.Schum.* | LC | Bats | Wind/ water | Tree |
| INPA Reserves, Brazil | *Ceiba jasminodora (A.St.-Hil.) K.Schum.* | VU | Moths | Wind/ water | Tree |
| INPA Reserves, Brazil | *Ceiba schottii Britten & Baker f.* | LC | Butterflies | Wind/ water | Tree |
| INPA Reserves, Brazil | *Ceiba trischistandra (A.Gray) Bakh.* | N/A | Bats, bees, hummingbirds | Wind | Tree |
| INPA Reserves, Brazil | *Peltogyne pauciflora Benth.* | LC | Bees | Wind | Tree |
| INPA Reserves, Brazil | *Peltogyne recifensis Ducke* | NT | Bees, bats, birds | Wind | Tree |
| INPA Reserves, Brazil | *Porcelia ponderosa Rusby* | LC | Beetles | Animal | Tree |
| Tropenbos International, Bolivia | *Bougainvillea modesta Heimerl* | LC | Bees, butterflies | Wind/ water | Tree |
| Tropenbos International, Bolivia | *Bougainvillea stipitata Griseb.* | N/A | Butterfly, moth | Wind/ water | Tree |
| Catimbau National Park, Brazil | *Aeschynomene martii Benth.* | N/A | Bee | Water | Shrub |
| Catimbau National Park, Brazil | *Barnebya harleyi W.R.Anderson & Gates* | LC | Bee (Frieseomelitta meadewaldoi) | Wind | Climbing tree |
| Catimbau National Park, Brazil | *Caesalpinia pluviosa DC.* | N/A | Butterflies, bees, hummingbirds | Wind/ water | Tree |
| Catimbau National Park, Brazil | *Mimosa acutistipula (Mart.) Benth.* | N/A | Wind, bees | Wind/ water | Tree/ shrub |
| Catimbau National Park, Brazil | *Pseudobombax parvifolium Carv.-Sobr. & L.P.Queiroz* | LC – one paper argues it should be NT | Bats, moths | Wind/ gravity | Tree |
| Catimbau National Park, Brazil | *Terminalia fagifolia Mart.* | LC | Bees, butterflies | Cassowaries | Tree |
| Cocha Cashu, Peru | *Dioscorea bulbifera L.* | N/A | Insects | Wind – bulbils | climbing herbaceous |
| Bia National Park, Ghana | *Aeschynomene indica L.* | LC | True bugs, wasp, wind, Ceratina japonica | Water | Herb |
| Bia National Park, Ghana | *Alafia barteri Oliv.* | N/A | Bees, butterflies | Mammals | Shrub |
| Bia National Park, Ghana | *Annona glauca Schumach. & Thonn.* | N/A | Beetles | Fish | Tree |
| Bia National Park, Ghana | *Combretum acutum M.A.Lawson* | N/A | Unknown | Wind | Tree |
| Bia National Park, Ghana | *Crotalaria mortonii Hepper* | LC | Bees | Wind/ water | Subshrub |
| Bia National Park, Ghana | *Dracaena phrynioides Hook.* | N/A | Hawk moth | Primates, birds, rodents | Subshrub |
| Bia National Park, Ghana | *Landolphia micrantha (A.Chev.) Pichon* | N/A | Bees, ants, butterflies | Primates | Vine |
| Bia National Park, Ghana | *Marantochloa leucantha (K. Schum.)* | N/A | Bees | Primates | Herb |
| Bia National Park, Ghana | *Stylosanthes erecta P.Beauv.* | N/A | Bees | Water, small animals | subshrub |
| Bia National Park, Ghana | *Terminalia laxiflora Engl.* | LC | Bees | Wind | Tree |
| Bia National Park, Ghana | *Tetrorchidium didymostemon (Baill.) Pax & K.Hoffm.* | LC | Insect | Unknown | Tree |

*(Continued)*

**Table 2.** (Continued)

| Location | Species | IUCN | Pollinator | Seed dispersal | Habit |
|---|---|---|---|---|---|
| Bia National Park, Ghana | *Tricalysia pallens Hiern.* | LC | Insect | Birds | Shrub |
| Bia National Park, Ghana | *Triclisia subcordata Oliv.* | N/A | Insect | Primates | Climbing |
| Bia National Park, Ghana | *Vangueriella nigerica (Robyns) Verdc.* | LC | Bees, butterflies, beetles, wind | Wind/ water | Tree |
| Southern Guinea Savanna Research Station, Guinea | *Diospyros lotus L.* | LC | Bees | Birds, mammals | Tree |
| Jatun Sacha, Ecuador | *Rudgea crassipetiolata Zappi & E.Lucas* | N/A | Bees | Wind, water, birds, mammals | Tree |
| Isthmus of Kra, Thailand/ Myanmar | *Nymphoides aurantiaca (Dalzell) Kuntze* | LC | Insect | Water | Aquatic |

measure of peak or first flowering cannot be guaranteed, thus we have decided to use mean flowering, as mean flowering or average flowering has been shown to be accurately represented by museum specimens as well as be biologically informative [25–30,32,33]. Additionally, it has been shown that phenology studies with field observations yielded similar results to those utilizing only herbarium specimens [25–30,32–34].

## Construction of datasets

Flowering data were collected from GBIF, a publicly available repository of natural history collections and observations; for purposes of this study, we retained data that derived only from museum specimens, specifically excluding other types of data (e.g., observations). Using the map polygon feature on GBIF, a polygon was drawn around the boarders of each preserve. A list of all angiosperm species on GBIF within the bounds of polygons, which approximated the bounds of each biological research stations/ preserve, was compiled (GBIF.org, January 23rd- April 17th 2025). Selecting only preserved specimens; flowering specimens were downloaded in complete Darwin Core format. From this initial list, we manually reviewed photos of digitized specimens to determine presence or absence of flowering. It is generally considered best practice for Herbarium specimens to be prepared with reproductive material, namely flower, fruit or ideally both. All flowering phenophases were included in this study, i.e., flowering was considered "present" if reproductive whorls were visible. Using the filter by month feature on GBIF, we recorded every month in which flowering was occurring, for each species, in each year, across all collections. A total of 8225 specimens were utilized in this study, with an average of 274 specimens per species (S1 Table). Of the species included in this study, 21 are trees, seven are shrubs or subshrubs, four are climbing, three are herbs and one is aquatic (Table 2). All species included in this study utilize animal pollinators, with most using a variety of insects and some using a vertebrates such as bats and birds. Two species also utilize wind pollination. Of the species included in this study, 22 species use abiotic seed dispersal, such as wind and water, and 11 use animal seed dispersal. According to the IUCN red list, 17 species are categorized as Least Concern, One species is categorized as Near Threatened (*Peltogyne recifensis*), and One species is categorized as Vulnerable (*Ceiba jasminodora*) (IUCN 2015). Of the species included in this study, the IUCN did not have a report on 15 species. The IUCN reports the population is stable for 12 species and decreasing for one species (*Espeletia brachyaxiantha*)(IUCN 2025). All other population changes are Unknown (Table 2).

## Data subsets

We constructed six datasets considered final for purposes of downstream exploration, following implementation of four criteria (Table 3) for inclusion of a given species in each matrix: (1) maximum number of months flowering, (2) minimum

**Table 3. Six datasets were constructed by implementing four different criteria that span four different parameters: (1) maximum number of months flowering, (2) minimum number of herbarium specimens per species, (3) minimum span of years collected in the flowering phenophase, and (4) maximum number of specimens collected in a single day.**

| Dataset | Months flowering | # Specimens | Year span | Specimens per day | # Species meeting criteria |
|---------|------------------|-------------|-----------|-------------------|----------------------------|
| 1 | ≤4 | ≥20 | ≥29 | Not Applicable | 33 |
| 2 | ≤3 | ≥20 | ≥29 | Not Applicable | 16 |
| 3 | ≤4 | >50 | ≥29 | Not Applicable | 26 |
| 4 | ≤4 | ≥20 | On or after 1960 | Not Applicable | 32 |
| 5 | ≤4 | ≥20 | Before 1960 | Not Applicable | 11 |
| 6 | ≤4 | ≥20 | ≥29 | Max 1 per day | 33 |

number of herbarium specimens, (3) minimum span of years collected, and (4) maximum number of specimens collected in a single day (Table 3).

## Statistical analyses

To facilitate the analysis of phenological shifts through time, we converted flowering date into Julian date and year. In this study, Julian date refers to the date of collection, as the date of first flowering or peak flowering cannot be determined from specimen records [25]. Utilizing collection dates has been shown to represent a reliable estimate of flowering period [28,35].

**Determining changes in flowering date.** To test changes in flowering date in all species in each of the six datasets, we conducted a series of analyses regressing Julian date of flowering onto year. These data have a circular (Von Mises) distribution owing to some species flowering periods spanning the new year [36–43]. We employed circular GLM statistics following Mulder and Klugkist (2017). Leap years were identified, and dates were scaled appropriately as a year consisting of 366 days. Julian dates were converted to radians using the formula (2 * pi * day-of-year/ 365 – pi). A Bayesian generalized linear model with a Von Mises distribution was run utilizing the circglmbayes package in R [44,45]. For a circular response y on -pi:pi and a single continuous predictor x, the model we used was a: $y = \beta_0 + inv\_tan\_half (\beta_1 * x)$. The training algorithm was a mix of Gibbs sampling, fast rejection sampling, and Metropolis-Hastings. Using two training algorithms is the most efficient and stable way to fit a Bayesian model where no single sampling regime works for all parameters. Four chains were used, with a burnin of 200 and 2500 iterations. The coefficients were saved to later create summary statistics. Separate regressions were run for each species, individually.

For each species, circular slope values were derived along with the associated standard deviation of the posterior samples for each slope parameter and date range. Uncertainty was assessed using standard deviations of the posterior samples for each slope parameter associated with each species. Utilizing flowering date values from the regression and span of years, changes in flowering date per decade (ΔDOY/decade) were calculated. Absolut value of the slopes were calculated to assess and compare magnitude of the change in flowering date irrespective of direction. Averages of the absolute value of the (circular) slope and ΔDOY/decade were calculated.

**Affiliation and directionality of shifts.** We summarized the counts of how many species had a positive slope (indicating flowering is now occurring later in the year vs. earlier time periods) compared to the number of species that had a negative slope (indicating flowering is now occurring earlier in the year, vs. earlier time periods).

## Phylogenetic signal

We assembled a molecular phylogeny for 33 tropical plant species using publicly available internal transcribed spacer (ITS) sequence data from GenBank. Species names were matched to NCBI taxonomy using the rentrez R package, and

the five most relevant ITS sequences per species were retrieved. Sequences were downloaded and written to a FASTA file for alignment.

Sequence alignment was performed using the ClustalW algorithm via the msa R package. The resulting multiple sequence alignment was exported in FASTA format and converted to DNAbin format for downstream analysis.

A maximum likelihood (ML) phylogeny was inferred using the phangorn package in R. First, a neighbor-joining (NJ) tree was constructed from a pairwise distance matrix and used as a starting tree. A General Time Reversible (GTR) model with estimates of invariant sites and among-site rate variation (G + I model) was applied. Model parameters and tree topology were optimized with stochastic rearrangement. Bootstrap support was calculated from 100 replicates and mapped onto the ML topology.

To assess phylogenetic signal in flowering phenology, we compiled species-level estimates of the change in flowering date per decade derived from regression slopes of flowering Julian day over time. Species names were matched to phylogenetic tip labels, and the tree was pruned to retain only taxa with available trait data.

We calculated Blomberg's K, a measure of phylogenetic signal using the phytools package, with 1,000 random permutations to assess statistical significance. Because the ML tree lacked branch lengths for some tips, we ensured all edges had positive length either by using compute.brlen() with the Grafen method or assigning a small constant value to near-zero branches.

## Results

### ΔDOY/decade

The average absolute ΔDOY/decade across all species was 2.04 days per decade. The species with the largest ΔDOY/decade was *Peltogyne recifensis* in the INPA Reserves Brazil, with a ΔDOY/decade of 14.1 days later per decade. This represents a total change in flowering date of 80.47 days later over 57 years, from 1951–2008. The largest ΔDOY/decade is an order of magnitude greater than the second largest ΔDOY/decade, which is more representative of the overall results, and therefore included here: The species with the second largest ΔDOY/decade is *Barnebya harleyi* in Catimbau National Park in Caatinga Brazil with a ΔDOY/decade of 5.84 days later per decade. This represents a total change in flowering date of 29.82 days later over 51 years, from 1971–2022. The smallest change in flowering later in the year was *Dioscorea bulbifera* in Cocha Cashu Peru with a ΔDOY/decade of 0.17 days later per decade. This represents a total change in flowering date of 3.98 days later over 229 years, from 1794–2023.

The species with the largest ΔDOY/decade earlier in the year was *Crotalaria mortonii* in Bia National Park Ghana with a ΔDOY/decade of 4.08 days earlier per decade. This represents a total change in flowering date of 17.15 days earlier over 42 years, from 1953–1995.The species with the smallest ΔDOY/decade was *Rudgea crassipetiolata* in Jatun Sacha Ecuador with a ΔDOY/decade of 0.0369 days earlier per decade. This represents a total change in flowering date of 0.17 days earlier over 46 years, from 1974–2020 (S1 Table; Fig 2). Of the 33 species, 10 had negative slopes, indicating a change in flowering to earlier in the year, whereas 23 species had a positive slope, indicating a change in flowering to later in the year (S1 Table; Fig 3). The impacts of each of the six data subsets can be seen in Table 4.

### Phylogenetic signal

We found a significant phylogenetic signal in the magnitude of change in flowering time across species (Blomberg's K = 0.266, $P$ = 0.001) (Fig 4). Although the value of K was substantially lower than 1, indicating that closely related species do not resemble each other as strongly as expected under a Brownian motion model of trait evolution, the signal was still stronger than would be expected by chance. This suggests that while flowering phenological shifts exhibit a weak overall tendency to be phylogenetically conserved, evolutionary history still plays a statistically significant role in explaining interspecific variation in flowering responses over time (Table 3).

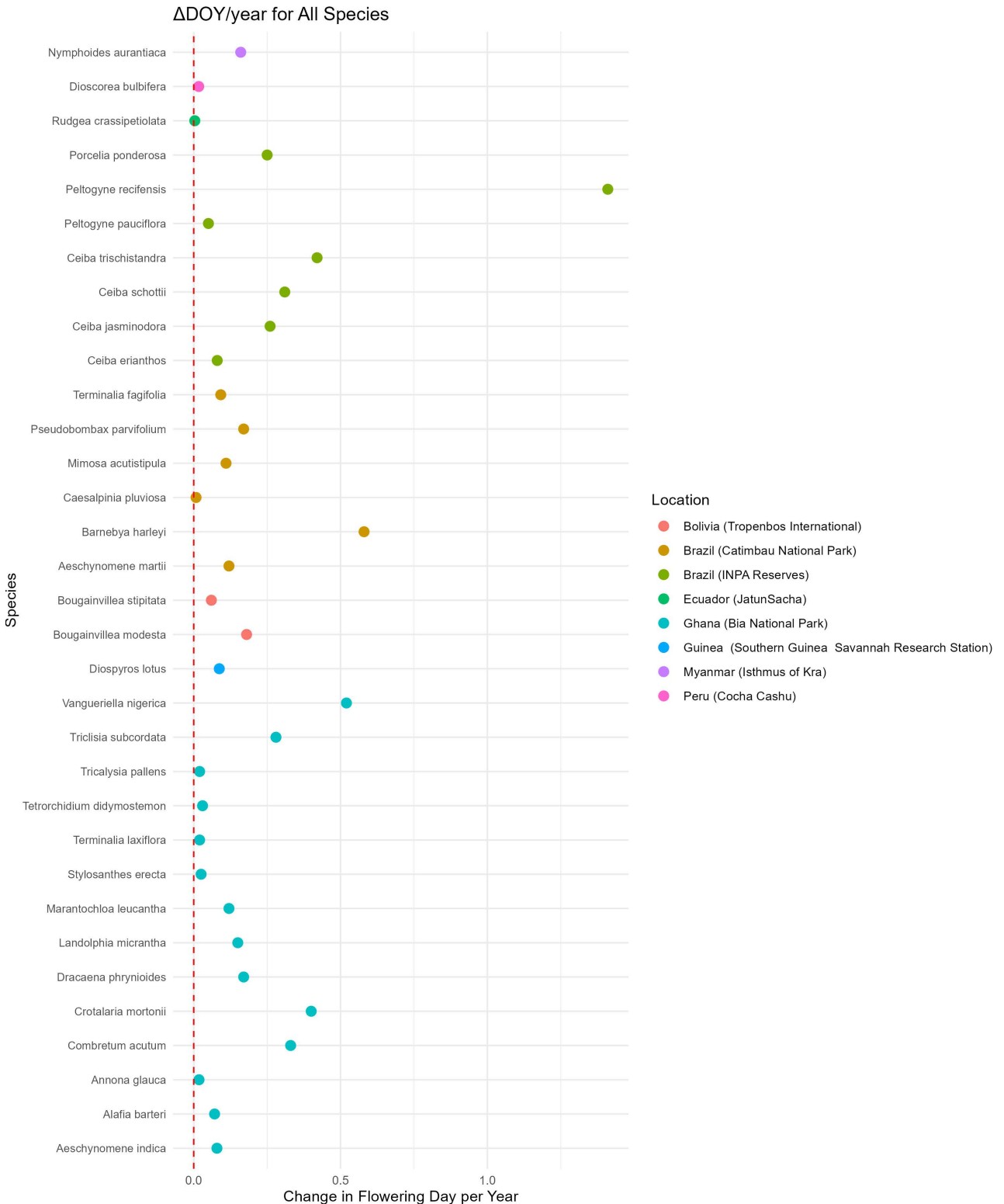

**Fig 2. Change in flowering date per decade of species that flower for 4 months or less, with a minimum of 20 specimens, spanning a minimum of 29 years.** Species arranged by location.

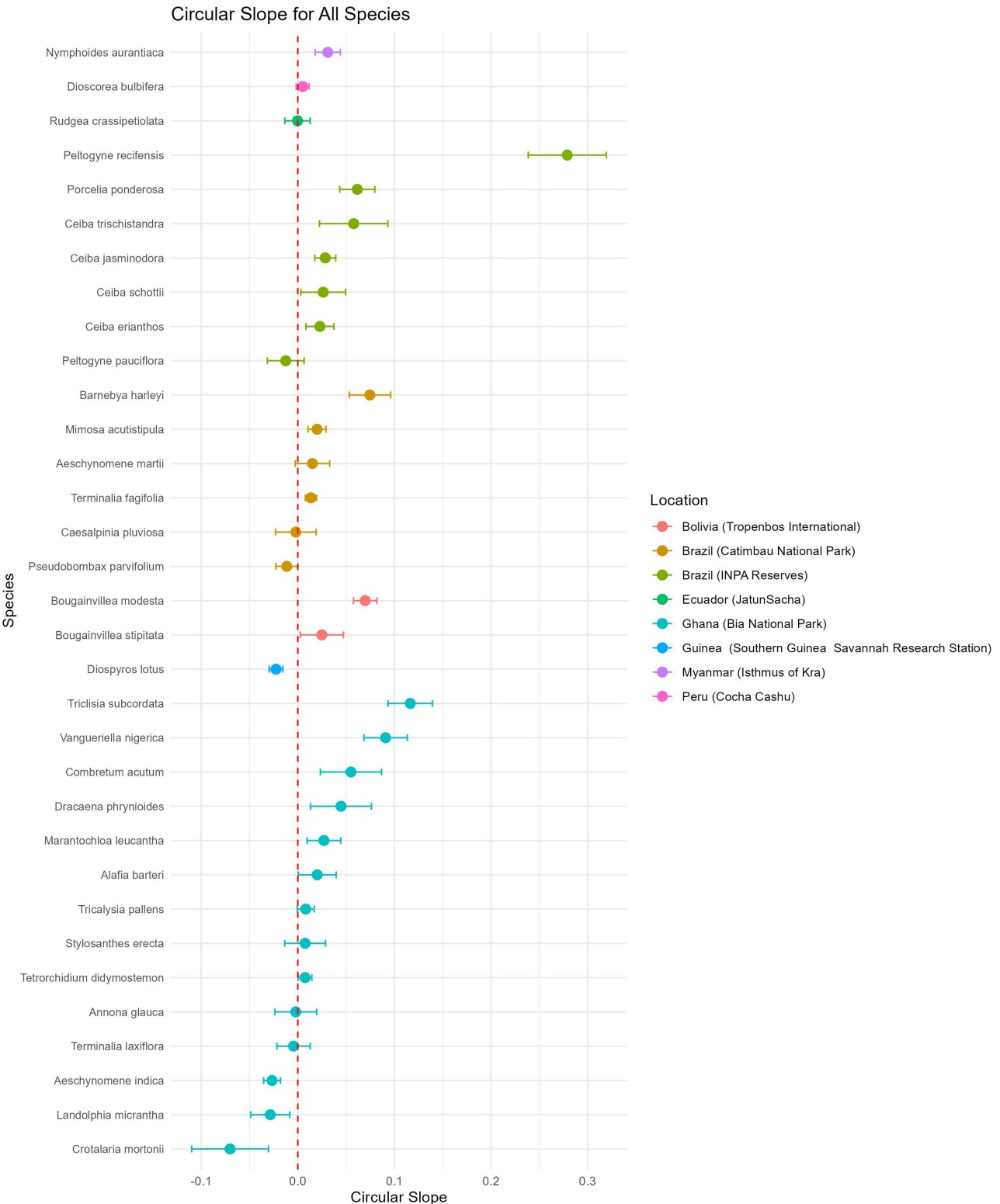

**Fig 3. Circular Slope and standard deviation of species that flower for 4 months or less, with a minimum of 20 specimens, spanning a minimum of 29 years.** Species arranged by location.

**Table 4. Averages pulled from each of the six datasets.** Dataset number, average absolute slope, SD, number of species with positive slope (indicating flowering occurring later in the year than historic records), number of species with negative slope (indicating flowering occurring earlier in the year than historic records), average change in flowering date per decade and number of species that met the dataset criteria (Table 3).

| Dataset | Slope | SD | Positive slope | Negative slope | Days per decade | # species |
|---|---|---|---|---|---|---|
| 1 | 0.028 | 0.04 | 23 | 10 | 2.03 | 33 |
| 2 | 0.033 | 0.015 | 11 | 5 | 1.63 | 16 |
| 3 | 0.026 | 0.016 | 17 | 7 | 1.47 | 26 |
| 4 | 0.036 | 0.02 | 20 | 12 | 2.35 | 32 |
| 5 | 0.043 | 0.028 | 6 | 4 | 2.14 | 10 |
| 6 | 0.021 | 0.032 | 23 | 9 | 1.85 | 33 |

## Discussion

Our research builds upon a growing body of evidence documenting impacts on the life cycles of plant species by bringing new information to light from tropical latitudes. Our work utilizing thousands of museum specimens not only demonstrated changes in flowering phenology of tropical plants but showed that the magnitude of change was comparable to those previously documented in non-tropical latitudes, therefore rejecting our hypothesis that tropical shifts are smaller than those seen in non-tropical latitudes. Our work furthermore showed a lack of uniformity in direction of shifts, consistent with our prediction that some species will flower earlier compared to historic records and others will flower later than historic records. These changes reflect crucial knowledge in our efforts to more completely understand the impacts of climate change by incorporating new data from tropical latitudes.

Across all 33 species in this study, we documented an average absolute change in flowering times of 2.04 ΔDOY/decade. This number reflects values previously derived from non-tropical studies. In a study that considered diverse biomes from across Earth, i.e., temperate, boreal, and tropical ecosystems, it has been shown that there has been an average phenological shift (ΔDOY/decade) of 2.51 days [46]. Other studies to have emphasized a regional perspective have shown greater or smaller shifts. For example, in British plants, Fitter & Fitter (2002) found an average shift in average flowering date of 4.5 days between 1991 and 2000. Auffret (2021) found that flowering start dates among plants in the Swedish flora shifted an average of days per decade. Similarly, Speed et al., (2022) found that Norwegian plant flowering shifted start date an average of 0.9 days per decade. Across Northern Canada, Bjorkman et al., (2015) found an average lengthening of flowering period of 1.2 days per decade. In alpine Tibet, Chen et al., (2023) found that flowering date advanced an average of 10.3 and 7.5 days respectively across two years of the study. In the southwestern United states, Rafferty et al., (2020) found shifts in average flowering duration ranged from 3.6 days per decade to 25 days per decade, from high to low elevation respectively. In Massachusetts, United States, Bertin (2015) found an average change in flowering onset of 2.9 days per decade. Across central North America, Austin et al. (2024) showed that species increased their flowering duration of 1.15 days per decade. In the southern hemisphere, in Sydney Australia, Everingham, et al., (2023) found that changes in mean flowering date ranged from 2.8 days per decade to 17.8 days per decade. Despite differences in drivers of flowering time found in tropical climates compared to non-tropical climates, our results show clearly that tropical latitudes have been comparably affected by climate change, and in some instances (e.g., northern boreal regions), shifts in tropical plants have been more severe. We emphasize that our results are based on species with discrete flowering periods, therefore removing artificial inflation that can result from consideration of species with longer "continuous" flowering periods [28,47–52].

### Directionality of change in flowering date

We found that roughly 1/3 of the species in our study (i.e., 10 of 33; Table 2, Fig 2) are now flowering earlier in the year compared to historical flowering date, and approximately 2/3 of the species (i.e., 23 of 33; Table 2, Fig 2) are flowering

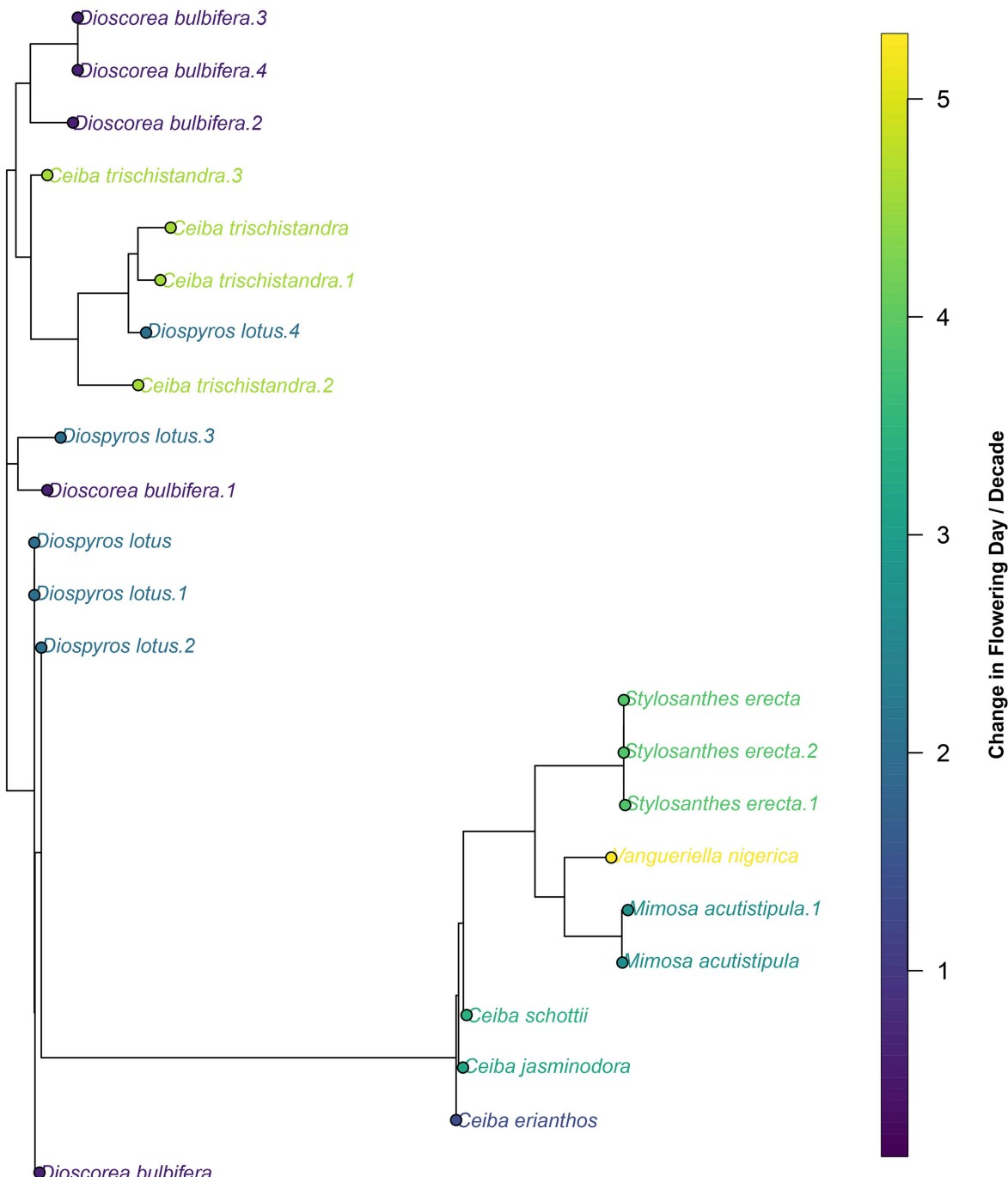

**Fig 4. Unrooted Maximum Likelihood phylogeny (inferred using the phangorn package) of 33 species included in this study, constructed from multiple representative sequences per each tip species.** Bootstrap support shown below branches where ≥ 70% (values less than 70 were omitted). Change in flowering date per decade was aligned to each species. We calculated Blomberg's K with 1,000 random permutations to assess statistical significance. Blomberg's K = 0.266, $P$ = 0.001.

later in the year compared to historical flowering date. We expected that tropical species flowering at comparable times in a given year may not all change their flowering in the same direction, and this pattern was supported [1–3]. Although there is not strict uniformity in the direction of change in flowering date in non-tropical locations, there is a greater consistency, due in large part to the drivers of flowering, such as increase or decrease in temperature and photoperiod [28,47–52]. The triggers of flowering in the tropics include myriad factors such as irradiance and solar availability [53–57], temperature [57–59], precipitation [57,60,61], and pollinator availability [57,62], thus some taxa may have separate flowering triggers than others occupying the same habitat. Excluding *Peltogyne recifensis*, we recovered comparable shifts in flowering days across the remainder of species (i.e., 32 of 33 species), with a total range of 0.369 to 5.84 days per decade (i.e., 5.81 days per decade) (S1 Table). In contrast, the number of days shifted in *Peltogyne recifensis* was order of magnitude greater than flowering changes in the remaining species. The reason for this difference remains unknown. Further studies focusing on the genus *Peltogyne* could provide further insights and yield possible conservation strategies.

## Biological impact

All the species included in this study had mutualistic interactions with at least one animal species. All species utilized animal pollinators. This reliance on animal pollination gives rise to the risk of misalignment with said pollinators. This misalignment can lead to changes in fitness for the species [5,6,9–14,16,17]. Of the species included in this study, 11 species utilize animals for seed dispersal. The change in reproductive timing of the plants can affect the abundance of the fruit available for consumption by the animals who rely on them [5,6,16,17]. Many of the animals who rely on these plants are primates, who are already an at-risk taxa (IUCN). The greater ecological implications of the changes in flowering time observed here cannot fully be determined, as we are looking at a small subset of the overall biodiversity found in the tropics, and the change in behavior of mutualistic partners cannot be determined here. Further exploration into the greater ecological changes seen should be a point of priority for the scientific community.

## Impacts of implementation of dataset criteria

In this study, we explored whether and how four different criteria involved in the construction of (six) different datasets impacted resultant patterns in tropical plant phenological shifts. Specifically, we considered the maximum number of total months flowering, the minimum number of herbarium specimens per species, the minimum span of years a species was collected in the flowering phenophase, and the maximum number of specimens (of a given species) collected from a single site in a single day (Tables 3 and 4). We found minimal variation in resultant change in flowering date across the (six) datasets. The largest variance from the overall dataset was in collections made after 1960, which had a marginally greater change in flowering date (2.35 days per decade) compared to the overall average (2.04 days per decade). This change is likely due to the increasing change in climate seen in recent decades [1–20].

We considered whether flowering duration had an impact on resultant days shifted [6,52] (Tables 3 and 4). Under the assumption that measures in flowering phenology are only valid for organisms that (1) only flower once per year and (2) have relatively short, cyclical flowering events, we aimed to provide an analysis of how flowering period length affected estimated flowering shift magnitude. Our analyses showed that shifts in number of days were comparable between analyses that considered whether flowering duration was less than 4 months vs. flowering duration less than 3 months) (Table 4). Future work should continue to explore this complex topic, particularly in portions of the planet where longer flowering periods are common, to assess whether our results are likely to be widespread, as others have found.

## Conclusion

We documented changes in flowering dates across 33 species of tropical flowering plants. We show that phenological change in tropical plants with discrete flowering times has been comparable to those seen in non-tropical environments. Furthermore, we have shown shifts have not been seasonally uniform across taxa, perhaps as a function of how triggers

of flowering differentially impact tropical plants compared to how they impact plants in non-tropical latitudes. What remains to be explored in tropical ecosystems is the degree to which abiotic factors such as temperature and precipitation, as well as biotic factors such as pollination and herbivory, impact flowering (Graves *et al.,* in prep).

## Supporting information

**S1 File. Suplimentary file contains additional information on data subsets used in this study.**
(DOCX)

**S1 Table. Location, plant species, plant family, slope of regression, regressing flowering date on year with Von Mises distribution, standard deviation, total change in flowering date scaled to change in days per decade, duration of flowering (in months), years said specimens spanned, and number of specimens.**
(DOCX)

**S1 Fig. Circular Slope and circular standard deviation of species that flower for 3 months or less, with a minimum of 20 specimens, spanning a minimum of 29 years (Dataset 2).** Species arranged by location.
(PDF)

**S2 Fig. ΔDOY/year of species that flower for 3 months or less, with a minimum of 20 specimens, spanning a minimum of 29 years (Dataset 2).** Species arranged by location.
(PDF)

**S3 Fig. Circular Slope and circular standard deviation of species that flower for 4 months or less, with a minimum of 50 specimens, spanning a minimum of 29 years (Dataset 3).** Species arranged by location.
(PDF)

**S4 Fig. ΔDOY/year of species that flower for 4 months or less, with a minimum of 50 specimens, spanning a minimum of 29 years (Dataset 3).** Species arranged by location.
(PDF)

**S5 Fig. Circular Slope and circular standard deviation of species that flower for 4 months or less, with a minimum of 20 specimens, only including specimens from 1960 or later (Dataset 4).** Species arranged by location.
(PDF)

**S6 Fig. ΔDOY/year of species that flower for 4 months or less, with a minimum of 20 specimens, only including specimens from 1960 or later (Dataset 4).** Species arranged by location.
(PDF)

**S7 Fig. Circular Slope and circular standard deviation of species that flower for 4 months or less, with a minimum of 20 specimens, Only including specimens from before 1960 (Dataset 5).** Species arranged by location.
(PDF)

**S8 Fig. ΔDOY/year of species that flower for 4 months or less, with a minimum of 20 specimens, Only including specimens from before 1960 (Dataset 5).** Species arranged by location.
(PDF)

**S9 Fig. Circular Slope and circular standard deviation of species that flower for 4 months or less, with a minimum of 20 specimens, spanning a minimum of 29 years – Only including one specimen per species per day (Dataset 6).** Species arranged by location.
(PDF)

**S10 Fig. ΔDOY/year of species that flower for 4 months or less, with a minimum of 20 specimens, spanning a minimum of 29 years – Only including one specimen per species per day (Dataset 6).** Species arranged by location. (PDF)

## Acknowledgments

The authors thank Ryan Hollingsworth for editorial support, Dani Bosse, Kendall Origer, Thummim Pradhan, Aurora Health, Anika Shoemaker, and Gerano Morales for help with scoring GBIF images to determine flowering periods, Seth Raynor for editorial support, Brett Melbourne for aid in statistical analyses and debugging, Beelzebub The Cat for contributions and support, and Gladiana Spitz for aid in code design and structure, debugging and design of figures.

## Author contributions

**Conceptualization:** Skylar Graves.

**Data curation:** Skylar Graves.

**Formal analysis:** Skylar Graves.

**Investigation:** Skylar Graves.

**Methodology:** Skylar Graves.

**Software:** Skylar Graves.

**Supervision:** Erin A. Manzitto-Tripp.

**Validation:** Skylar Graves.

**Visualization:** Skylar Graves.

**Writing – original draft:** Skylar Graves.

**Writing – review & editing:** Skylar Graves, Erin A. Manzitto-Tripp.

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

**59.** Nepstad DC, Tohver IM, Ray D, Moutinho P, Cardinot G. Mortality of large trees and lianas following experimental drought in an Amazon forest. Ecology. 2007;88(9):2259–69. https://doi.org/10.1890/06-1046.1 PMID: 17918404

**60.** Janzen DH. Why Mountain Passes are Higher in the Tropics. Am Natural. 1967;101(919):233–49. https://doi.org/10.1086/282487

**61.** Phillips OL, Aragão LEOC, Lewis SL, Fisher JB, Lloyd J, López-González G, et al. Drought sensitivity of the Amazon rainforest. Science. 2009;323(5919):1344–7. https://doi.org/10.1126/science.1164033 PMID: 19265020

**62.** Morellato LPC, Alberton B, Alvarado ST, Borges B, Buisson E, Camargo MGG, et al. Linking plant phenology to conservation biology. Biol Conserv. 2016;195:60–72. https://doi.org/10.1016/j.biocon.2015.12.033

