## [Decision Letter · Decision Letter 0]

9 Dec 2025

Dear Dr. Graves,

Thank you for submitting your manuscript to PLOS ONE. After careful consideration, we feel that it has merit but does not fully meet PLOS ONE’s publication criteria as it currently stands. Therefore, we invite you to submit a revised version of the manuscript that addresses the points raised during the review process.

We look forward to receiving your revised manuscript.

Kind regards,

Hong Qin, PhD

Academic Editor

PLOS One

Journal Requirements:

2. We note that Figure 1 in your submission contains a map image which may be copyrighted. All PLOS content is published under the Creative Commons Attribution License (CC BY 4.0), which means that the manuscript, images, and Supporting Information files will be freely available online, and any third party is permitted to access, download, copy, distribute, and use these materials in any way, even commercially, with proper attribution. For these reasons, we cannot publish previously copyrighted maps or satellite images created using proprietary data, such as Google software (Google Maps, Street View, and Earth). For more information, see our copyright guidelines: http://journals.plos.org/plosone/s/licenses-and-copyright.

3. Please include captions for your Supporting Information files at the end of your manuscript, and update any in-text citations to match accordingly. Please see our Supporting Information guidelines for more information: http://journals.plos.org/plosone/s/supporting-information .

4. Please upload a new copy of Figure S6 as the detail is not clear. Please follow the link for more information:  https://journals.plos.org/plosone/s/figures

Reviewers' comments:

Reviewer's Responses to Questions

**Comments to the Author**

1. Is the manuscript technically sound, and do the data support the conclusions?

Reviewer #1: Partly

Reviewer #2: Yes

2. Has the statistical analysis been performed appropriately and rigorously?

Reviewer #1: No

Reviewer #2: Yes

3. Have the authors made all data underlying the findings in their manuscript fully available?

Reviewer #1: Yes

Reviewer #2: Yes

4. Is the manuscript presented in an intelligible fashion and written in standard English?

Reviewer #1: Yes

Reviewer #2: Yes

Reviewer #1: Details about responses to the questions above: Some of the data do not support the conclusions, particularly in the comparisons of the findings regarding shifts in flowering phenology in tropical latitudes to findings from other locations. Specifically, more attention needs to be given to the direction of shifts, the uniformity of shift directions among species, and the 'severity' of responses. Given that collection time is proxying for flowering time, attempts to tease out any effects of collection time on the flowering phenological results could be made or explanations could be give for why this may not be possible. The fit of the regression lines used to determine the responses of flowering to time should be analyzed. The criteria for inclusion of specimens in the study should be better justified.

Additional comments:

Abstract

More justification for the potential ecological significance of the study focus rather than justification based solely on the lack of previous studies with this focus would strengthen the overall presentation of the work.

The second bullet point cites ‘this hypothesis’. To emphasize the reference, I would suggest changing ‘postulated’ to ‘hypothesized’ in the previous bullet point. Also in the second bullet point, an ‘of’ is missing between ‘phenology flowering’.

Use consistency in the presentation of significant digits in the ‘Results’.

More ecological context in the ‘Conclusion’ section would strengthen the Abstract.

Introduction

Is ‘anthropogenic-mediated’ relevant in the first sentence? Such shifts would apply to climate change more broadly. Also, this sentence needs citation.

The ‘cascade effects’ (line 34) could be extrapolated. Cascading impacts on what? I would rewrite this and the following sentence as one: ‘Responses of plants to climate change can have cascading impacts across ecosystems.’

The paragraph of lines 40-44 is very short. Could this be further described on combined with the previous paragraph?

The ‘universally recognized importance’ cited in line 45 needs support.

Some link between the biodiversity information in lines 46-49 and the study focus on phenology should be made. Does this biodiversity suggest the potential for strong cascading effects, for example?

What do flowers from plants with discrete flowering periods in the tropics generally respond to if not temperature- or photoperiod-dependent? If not temperature, why would flowering phenology shifts be hypothesized? What proportion of plant species in the tropics have discrete flowering periods? Is this common?

The sentence in lines 54-55 is unnecessary.

The paragraph in lines 60-63 is very short. Either elaborate or combine this with the previous paragraph as the topic seems the same. Also, the sentence in lines 60-61 seems overly obvious.

What are examples of global impacts that changes in the flowering phenology of tropical plants could have (lines 65-67)?

Were the 33 plants selected at random? Or were these the only ones that fit the discrete-flowering criterion for which specimens were available? If the latter, I would use the word ‘identified’ instead.

The paragraph in lines 71-77 is unnecessary. Limitations of the study should be described in the Discussion section.

Methods

Remove the first part of the first sentence unless it is made less general.

Was any analysis of collection efforts through time made to tease out any influence of collection effort changes on flowering phenology shifts? For example, was there no clear pattern of changes in collection efforts?

Herbarium collections ‘prior’ to when? (line 82) Also, I would change ‘regularity’ to ‘consistency’.

What ‘criteria for inclusion’ are being referenced in line 85?

‘Mean’ flowering vs. ‘peak’ flowering could be explained in lines 95-96. What was measured in the present study?

The statement in lines 96-98 is attributable to a single study and this should be described. This is not necessarily a broader generalization.

Is the first ‘species’ that appears in line 111 meant to be ‘specimens’?

Why was 4 or less months/year of flowering considered the cut-off for including species in this study? Why not 5 months or less? Would this have affected the results potentially? Similarly, please justify other inclusion criteria (i.e., 20 specimens per site, 29 years of data, etc.).

Was there any possibility of making distinctions between flowering stages that could have further informed the timing of flowering phenology?

Can some explanation of how collection and flowering generally align be added to strengthen the use of collection time as a proxy for flowering time? Flowering specimens are encouraged in herbarium collections.

Were line regressions used to derive the absolute slopes analyzed for significance (fit of the yearly points along the best-fit line)?

Results

Changes in flowering date should be described in terms of earlier vs. later? A change in and of itself is not as informative. I would present results for the negative sloped species distinctly from the positively sloped species as these could represent distinct ecological responses that could have distinct ecological relevance.

The figure captions should be more ‘stand-alone’ by explaining abbreviations used, study focus on the tropics, etc.

In the figures, could species be presented by family vs. location?

Suggest moving Table 2 to the Supplement. Some of the information in this table is double depicted in the figures.

Discussion

The study brings new ‘information’ to light, not new ‘data’.

In non-tropical latitudes, are shifts more likely to be in one direction (i.e., earlier in the spring, later in the fall)? Or do you similar see changes in both directions depending on the species and location?

The assertion that tropical shifts are not ‘less severe’ than those documented at other latitudes needs to be substantiated?

How ‘severe’ were those shifts? The existence of shifts and their severity are distinct concepts (lines 173-177).

Why was your prediction that the species would show a lack of uniformity in the direction of their flowering phenology shifts (line 177, lines 196-198)?

Lines 182-192: Again, the direction of the shifts seems ecological relevant. Direction should be discussed.

If precipitation is suspected to be a major driver of phenological shifts in the tropics, could that have been investigated here?

What major changes in precipitation have occurred through time in the study locations? Could that explain phenological shifts in flowering?

A major limitation of the Discussion is the lack of attention given to the ecological relevance of the findings. What could the detected shifts (which were not uniform in direction) mean for other species, communities, ecosystem? How pollinator-specific are these species (i.e., are they generalists or specialists in this regard)? Are they all primary animal-pollinated? Have other systems at other latitudes also showed such non-uniformity in the direction of phenological shifts, and if so, what has that meant for other species, communities, ecosystems?

The ’Human Impacts’ subsection seems extraneous. More important would be discussion of the potential ecological relevance of the findings.

Reviewer #2: In PONE-D-25-54215 entitled “Observing shifts in phenology of tropical flowering plants”, the authors test for phenological shifts in tropical flowering plants. They were particluarlly interested in comparing their results with temperate species and thus restricted their analyses to species with distinct seasonal patterns (i.e., blooming for 3-4 months). The manuscript was very easy to read and addressed a timely topic that would be of interest to a broad audience. Many of the species observed exhibited phenological shifts to those documented for temperate species, which has interesting ecological implications. I commend the authors for their significant efforts to ensure robust results and to discuss them without hyperbole. However, there are several parts of the manuscript that would benefit from increased clarity and depth (see my comments below for details). Upon revision, I think this manuscript will make an important contribution to the field.

While easy to read, the manuscript appeared to lack sufficient support for the claims upon first reading. However, this was because much of the pertinent information is in the supplementary materials and never referenced in the main text. The authors conducted several analyses to test potential biases or weaknesses in their analyses. These additional analyses indicate that their results are robust to those biases, but they were not described or referenced in the main text as they should be (indeed it would strengthen the paper to include them).

Moreover, two results deserve additional discussion. The first is the similar pacing of phenological change before and after 1960. Given that climate change is accelerating, seeing similar changes in both time periods could indicate a slower rate of response or constraints on phenological shifts. While the authors do not directly test that idea, it warrants further discussion in the main text. The presence of a phylogenetic signal, while not particularly strong, also deserves discussion in the main text.

As is common in phenology studies, species varied in the degree of change through time. However, the authors provide little rationale for why we might see such differences in this data set. What are the documented climate change patterns for these locations? What habitats are represented? In which habitats are the species found and do they specialize on specific habitats? Are these long-lived or short-lived species? What kinds of growth habits do they exhibit (e.g., forbs, woody vines or shrubs, trees)? All of these factors could influence species' response to climate changes, and it is unclear what subset of tropical habits were studied here. I am not suggesting that the authors have to change their analyses to test whether any of these factors predict phenological changes. However, these traits do warrant description in the methods and, if there are indicators that any of these factors might influence species responses, they should be addressed in the discussion.

Finally, the methods need more detail for them to be interpretable, particularly for those less familiar with circular linear regression (e.g., description of the purpose of the training algorithm and the various sampling regimes).

**Do you want your identity to be public for this peer review?** For information about this choice, including consent withdrawal, please see our Privacy Policy

Reviewer #1: No

Reviewer #2: No

---

## [Author Response · Author response to Decision Letter 1]

15 Dec 2025

Below is the same as the "Response to Reviewers" document:

We thank the editor and reviewers for taking the time to read and review our submission. We hope we have addressed your concerns thoroughly. Our response is in red.

Journal Requirements:

Style requirements changed for “Manuscript” file.

2. We note that Figure 1 in your submission contains a map image which may be copyrighted. All PLOS content is published under the Creative Commons Attribution License (CC BY 4.0), which means that the manuscript, images, and Supporting Information files will be freely available online, and any third party is permitted to access, download, copy, distribute, and use these materials in any way, even commercially, with proper attribution. For these reasons, we cannot publish previously copyrighted maps or satellite images created using proprietary data, such as Google software (Google Maps, Street View, and Earth). For more information, see our copyright guidelines: http://journals.plos.org/plosone/s/licenses-and-copyright.

New (non-copyright) map created using R package “maps”.

Most supporting information moved to main text. Remaining supporting information updated accordingly.

4. Please upload a new copy of Figure S6 as the detail is not clear. Please follow the link for more information: https://journals.plos.org/plosone/s/figures

New copy of Figure (now figure 9) uploaded

Reviewers' comments:

Reviewer's Responses to Questions

Comments to the Author

1. Is the manuscript technically sound, and do the data support the conclusions?

Reviewer #1: Partly

Reviewer #2: Yes

2. Has the statistical analysis been performed appropriately and rigorously?

Reviewer #1: No

Reviewer #2: Yes

3. Have the authors made all data underlying the findings in their manuscript fully available?

Reviewer #1: Yes

Reviewer #2: Yes

4. Is the manuscript presented in an intelligible fashion and written in standard English?

Reviewer #1: Yes

Reviewer #2: Yes

5. Review Comments to the Author

Reviewer #1: Details about responses to the questions above: Some of the data do not support the conclusions, particularly in the comparisons of the findings regarding shifts in flowering phenology in tropical latitudes to findings from other locations. Specifically, more attention needs to be given to the direction of shifts, the uniformity of shift directions among species, and the 'severity' of responses. Given that collection time is proxying for flowering time, attempts to tease out any effects of collection time on the flowering phenological results could be made or explanations could be give for why this may not be possible. The fit of the regression lines used to determine the responses of flowering to time should be analyzed. The criteria for inclusion of specimens in the study should be better justified.

Thank you for your feedback. The inclusion of the tests conducted (previously in the supplemental materials, now in the main text) should address the above concerns. Additional exposition of comparison to non-tropical latitudes added in discussion section. Direction of shifts specified in results and discussion. It has been shown that the unwanted impacts of collector bias can be ameliorated with large datasets and utilization of mean flowering, as opposed to peak or first flowering (25-30 32, 33-34). Additionally, it has been shown that phenology studies with field observations yielded similar results to those utilizing only herbarium specimens (25-30 32, 33-34). Due to the nature of a Bayesian analysis, testing the fit of the regression is built into the process. The training algorithm was a mix of Gibbs sampling, fast rejection sampling, and Metropolis-Hastings. Using two training algorithms is the most efficient and stable way to fit a Bayesian model where no single sampling regime works for all parameters. Four chains were used, with a burnin of 200 and 2500 iterations. The convergence of the chains indicates the best model fit. Furthermore, uncertainty was assessed using standard deviations of the posterior samples for each slope parameter associated with each species. The inclusion of the six data subsets should address the concern regarding the justification for the inclusion of the species.

Abstract

More justification for the potential ecological significance of the study focus rather than justification based solely on the lack of previous studies with this focus would strengthen the overall presentation of the work.

Understood. This change has been made.

The second bullet point cites ‘this hypothesis’. To emphasize the reference, I would suggest changing ‘postulated’ to ‘hypothesized’ in the previous bullet point. Also in the second bullet point, an ‘of’ is missing between ‘phenology flowering’.

Understood. This change has been made.

Use consistency in the presentation of significant digits in the ‘Results’.

Understood. This change has been made.

More ecological context in the ‘Conclusion’ section would strengthen the Abstract.

Understood. This change has been made.

Introduction

Is ‘anthropogenic-mediated’ relevant in the first sentence? Such shifts would apply to climate change more broadly. Also, this sentence needs citation.

Term ‘anthropogenic-mediated’ has been removed and citation has been added.

The ‘cascade effects’ (line 34) could be extrapolated. Cascading impacts on what? I would rewrite this and the following sentence as one: ‘Responses of plants to climate change can have cascading impacts across ecosystems.’

Extrapolation on the cascade effects has been done. Sentences have been re-written as one, as suggested.

The paragraph of lines 40-44 is very short. Could this be further described on combined with the previous paragraph?

Lines have been combined with previous paragraph.

The ‘universally recognized importance’ cited in line 45 needs support.

Citation added.

Some link between the biodiversity information in lines 46-49 and the study focus on phenology should be made. Does this biodiversity suggest the potential for strong cascading effects, for example?

Connecting information added.

What do flowers from plants with discrete flowering periods in the tropics generally respond to if not temperature- or photoperiod-dependent? If not temperature, why would flowering phenology shifts be hypothesized? What proportion of plant species in the tropics have discrete flowering periods? Is this common?

It is not that plants in the tropics don’t respond to temperature, it is that there is not the same distinct seasonality as a cold/dark induced dormancy period that cause either spring or autumn flowering to be most common across taxa. There is more complexity to the triggers of flowering in tropical latitudes. We address this topic in depth in a paper currently in review at PNAS.

The proportion of tropical plants with discrete flowering periods is relatively small, however, species that have discrete flowering periods are more tractable from a perspective of understanding phenological change, as these flowering patterns mirror those seen in non-tropical latitudes. Such species therefore enable comparison to phenological changes documented from non-tropical latitudes. From an ecological perspective, species with discrete flowering periods have a higher likelihood of misaligned interspecific interactions as a result of changes in reproductive period (22, 23). Measuring species with discrete flowering times is therefore likely to target the species most vulnerable to change.

The sentence in lines 54-55 is unnecessary.

Understood. This change has been made.

The paragraph in lines 60-63 is very short. Either elaborate or combine this with the previous paragraph as the topic seems the same. Also, the sentence in lines 60-61 seems overly obvious.

Understood. This change has been made.

What are examples of global impacts that changes in the flowering phenology of tropical plants could have (lines 65-67)?

Examples have been added.

Were the 33 plants selected at random? Or were these the only ones that fit the discrete-flowering criterion for which specimens were available? If the latter, I would use the word ‘identified’ instead.

These were the only ones that fit the discrete-flowering criterion for which specimens were available. Term changed to ‘identified’

The paragraph in lines 71-77 is unnecessary. Limitations of the study should be described in the Discussion section.

Paragraph deleted.

Methods

Remove the first part of the first sentence unless it is made less general.

First part of sentence removed.

Was any analysis of collection efforts through time made to tease out any influence of collection effort changes on flowering phenology shifts? For example, was there no clear pattern of changes in collection efforts?

Rates of collections were analyzed in selection of locations. With the exception of a reduction of collections during the World Wars, there was a steady rate of collections across time, hence the selection of these locations over ones with less frequent/regular collections.

Herbarium collections ‘prior’ to when? (line 82) Also, I would change ‘regularity’ to ‘consistency’.

Terminology changed as requested.

What ‘criteria for inclusion’ are being referenced in line 85?

Please see table 2 and all information moved from supplemental to main text.

‘Mean’ flowering vs. ‘peak’ flowering could be explained in lines 95-96. What was measured in the present study?

Explanation on difference between mean flowering, peak flowering and first flowering added. Mean flowering was used in this study.

The statement in lines 96-98 is attributable to a single study and this should be described. This is not necessarily a broader generalization.

I cited the first study to establish this idea, however I have now since added more studies that support this idea.

Is the first ‘species’ that appears in line 111 meant to be ‘specimens’?

Yes. Apologies for the typo.

Why was 4 or less months/year of flowering considered the cut-off for including species in this study? Why not 5 months or less? Would this have affected the results potentially? Similarly, please justify other inclusion criteria (i.e., 20 specimens per site, 29 years of data, etc.).

All of this was addressed in the supplemental, which has since been moved to the main text. We hope this explains the rationality effectively.

Was there any possibility of making distinctions between flowering stages that could have further informed the timing of flowering phenology?

It has been shown that the unwanted impacts of collector bias can be ameliorated with large datasets and utilization of mean flowering, as opposed to peak or first flowering (25-30 32, 33-34). For this reason, we chose to use Mean flowering. Although we could theoretically look at various phenophases, we would have less confidence in the results, and it would take additional mon

---

## [Decision Letter · Decision Letter 1]

11 Jan 2026

Dear Dr. Graves,

Thank you for submitting your manuscript to PLOS ONE. After careful consideration, we feel that it has merit but does not fully meet PLOS ONE’s publication criteria as it currently stands. Therefore, we invite you to submit a revised version of the manuscript that addresses the points raised during the review process.

We look forward to receiving your revised manuscript.

Kind regards,

Hong Qin, PhD

Academic Editor

PLOS One

**Journal Requirements:**

Reviewers' comments:

Reviewer's Responses to Questions

**Comments to the Author**

Reviewer #1: All comments have been addressed

Reviewer #2: All comments have been addressed

2. Is the manuscript technically sound, and do the data support the conclusions?

Reviewer #1: Yes

Reviewer #2: Yes

3. Has the statistical analysis been performed appropriately and rigorously?

Reviewer #1: Yes

Reviewer #2: Yes

4. Have the authors made all data underlying the findings in their manuscript fully available?

Reviewer #1: Yes

Reviewer #2: No

5. Is the manuscript presented in an intelligible fashion and written in standard English?

Reviewer #1: Yes

Reviewer #2: Yes

Reviewer #1: (No Response)

Reviewer #2: The author thoroughly responded to my recommendations (at times may have even over-corrected). I appreciate all of the changes the authors made to the text and tables but suggest that some of the added figures are unnecessary. Presenting the results of the competing models in table 3 was sufficient to address my concerns regarding providing evidence to support their claims in the text. In my opinion, Figures 4-8 could remain in the supplementary materials.

I would be interested in seeing the phylogeny with the change in flowering mapped on (color-coded by slope or similar would be sufficient). At present, the reader must cross reference multiple figures in order to connect the phylogeny in the figure with the stated results. If the response was mapped onto the phylogeny, the phylogenetic patterns would be much more interpretable.

Because the figure panels were each uploaded separately, the scaling of the text will be off once the panels are combined into a single figure, making the labels challenging to read. The authors will likely need to modify the figures to make them more legible.

The “Outlier Species” section should be included in the “Directionality of change...” section. In my opinion, it doesn’t warrant its own section.

Discussing how these analyses were generally robust to potential biases would be informative for the field. However, the length of the section “Impacts of implementation of dataset criteria” could be shortened to include only the first 2 paragraphs which synthesize the patterns rather than going through each criterion in detail.

Overall, I commend the authors for their thorough changes and look forward to seeing the final product in print.

**Do you want your identity to be public for this peer review?** For information about this choice, including consent withdrawal, please see our Privacy Policy

Reviewer #1: No

Reviewer #2: No

---

## [Author Response · Author response to Decision Letter 2]

15 Jan 2026

Below are the reviewers comments in red and my responses in black.

The author thoroughly responded to my recommendations (at times may have even over-corrected). I appreciate all of the changes the authors made to the text and tables but suggest that some of the added figures are unnecessary. Presenting the results of the competing models in table 3 was sufficient to address my concerns regarding providing evidence to support their claims in the text. In my opinion, Figures 4-8 could remain in the supplementary materials.

I am glad you found the corrections thorough! Figures 4-8 have been returned to the supplementary materials.

I would be interested in seeing the phylogeny with the change in flowering mapped on (color-coded by slope or similar would be sufficient). At present, the reader must cross reference multiple figures in order to connect the phylogeny in the figure with the stated results. If the response was mapped onto the phylogeny, the phylogenetic patterns would be much more interpretable.

I made a phylogeny color coded by the change in flowering date in days.

Because the figure panels were each uploaded separately, the scaling of the text will be off once the panels are combined into a single figure, making the labels challenging to read. The authors will likely need to modify the figures to make them more legible.

I believe this is referring to the 2 paneled figures of figures 4-8. Because these have been moved to the supplement, I simplified by making each figure independent instead of paneled. I split the figure captions to match.

The “Outlier Species” section should be included in the “Directionality of change...” section. In my opinion, it doesn’t warrant its own section.

I moved the “Outlier Species” section to the “Directionality of change” section

Discussing how these analyses were generally robust to potential biases would be informative for the field. However, the length of the section “Impacts of implementation of dataset criteria” could be shortened to include only the first 2 paragraphs which synthesize the patterns rather than going through each criterion in detail.

I shortened the section to only include the first two paragraphs as suggested.

Overall, I commend the authors for their thorough changes and look forward to seeing the final product in print.

Thank you for all your comments and suggestions. I really appreciate the attention to detail you put in to make this paper the best it can be!

---

## [Decision Letter · Decision Letter 2]

19 Jan 2026

Observing shifts in phenology of tropical flowering plants

PONE-D-25-54215R2

Dear Dr. Graves,

We’re pleased to inform you that your manuscript has been judged scientifically suitable for publication and will be formally accepted for publication once it meets all outstanding technical requirements.

Kind regards,

Hong Qin, PhD

Academic Editor

PLOS One

Additional Editor Comments (optional):

Reviewers' comments:

Reviewer's Responses to Questions

**Comments to the Author**

Reviewer #2: All comments have been addressed

2. Is the manuscript technically sound, and do the data support the conclusions?

Reviewer #2: Yes

3. Has the statistical analysis been performed appropriately and rigorously?

Reviewer #2: Yes

4. Have the authors made all data underlying the findings in their manuscript fully available?

Reviewer #2: Yes

5. Is the manuscript presented in an intelligible fashion and written in standard English?

Reviewer #2: Yes

Reviewer #2: The authors addressed all of my recommendations, and I believe the article is ready for acceptance. I look forward to seeing it in print!

**Do you want your identity to be public for this peer review?** For information about this choice, including consent withdrawal, please see our Privacy Policy

Reviewer #2: No

---

## [Editor Report · Acceptance letter]

PONE-D-25-54215R2

PLOS One

Dear Dr. Graves,

I'm pleased to inform you that your manuscript has been deemed suitable for publication in PLOS One. Congratulations! Your manuscript is now being handed over to our production team.

Kind regards,

on behalf of

Dr. Hong Qin

Academic Editor

PLOS One